# Subcellular Feature-Based Classification of α and β Cells Using Soft X-ray Tomography

**DOI:** 10.3390/cells13100869

**Published:** 2024-05-18

**Authors:** Aneesh Deshmukh, Kevin Chang, Janielle Cuala, Bieke Vanslembrouck, Senta Georgia, Valentina Loconte, Kate L. White

**Affiliations:** 1Department of Chemistry, Bridge Institute, Michelson Center for Convergent Bioscience, University of Southern California, Los Angeles, CA 90089, USA; aneeshde@usc.edu (A.D.); kchang42@usc.edu (K.C.);; 2Medical Biophysics Program, Keck School of Medicine, University of Southern California, Los Angeles, CA 90033, USA; 3Department of Anatomy, School of Medicine, University of California San Francisco, San Francisco, CA 94143, USA; 4Molecular Biophysics and Integrated Bioimaging Division, Lawrence Berkeley National Laboratory, Berkeley, CA 94720, USA; 5Department of Stem Cell Biology and Regenerative Medicine, Keck School of Medicine, University of Southern California, Los Angeles, CA 90033, USA

**Keywords:** soft X-ray tomography, cryogenic fluorescence microscopy, pancreatic islets, α cells, β cells, 3D cell mapping, machine learning, Uniform Manifold Approximation and Projection (UMAP)

## Abstract

The dysfunction of α and β cells in pancreatic islets can lead to diabetes. Many questions remain on the subcellular organization of islet cells during the progression of disease. Existing three-dimensional cellular mapping approaches face challenges such as time-intensive sample sectioning and subjective cellular identification. To address these challenges, we have developed a subcellular feature-based classification approach, which allows us to identify α and β cells and quantify their subcellular structural characteristics using soft X-ray tomography (SXT). We observed significant differences in whole-cell morphological and organelle statistics between the two cell types. Additionally, we characterize subtle biophysical differences between individual insulin and glucagon vesicles by analyzing vesicle size and molecular density distributions, which were not previously possible using other methods. These sub-vesicular parameters enable us to predict cell types systematically using supervised machine learning. We also visualize distinct vesicle and cell subtypes using Uniform Manifold Approximation and Projection (UMAP) embeddings, which provides us with an innovative approach to explore structural heterogeneity in islet cells. This methodology presents an innovative approach for tracking biologically meaningful heterogeneity in cells that can be applied to any cellular system.

## 1. Introduction

The islets of Langerhans are regulatory systems in the pancreas responsible for maintaining glucose homeostasis. Islets are primarily composed of four endocrine cell types: α, β, δ, ε, and pancreatic polypeptide (PP) cells, which secrete glucagon, insulin, somatostatin, ghrelin, and PP, respectively. β cells are the most abundant cell type and account for >70% of the cells in mouse islets. In comparison, α cells account for 10–20% of the total islet cell count, while δ, ε, and PP cells comprise less than 5% of islet cells [1,2]. Multiple fluorescence imaging modalities have described islet activity and crosstalk between cell types [3,4,5], which is important for glucose homeostasis [6,7].

Significant progress has been made in mapping primary murine α and β cells in large-scale 2D and 3D datasets using electron microscopy (EM), which provides high-resolution details of single cells and remarkable insights into subcellular structural organization [8,9,10,11]. However, EM techniques require fixation, staining, dehydration, and sectioning of the cell, which makes the technique labor-intensive. Additionally, previous studies have been limited to qualitative morphological features to differentiate cell types [11,12,13]. While cell and vesicle appearance are useful cues, their qualitative description lacks rigor and reproducibility. Thus, we require a more robust method to classify cell types that is also compatible with 3D cell mapping techniques. This would allow us to quantitatively capture the structural heterogeneity of cells while also retaining information about their identity. Ultimately, such a method would enable us to explore complex cellular rearrangements and dysfunction in α and β cells in disease.

Here, we use soft X-ray tomography (SXT) to enable high-resolution imaging of whole and fully hydrated cells (1–20 μm) [14,15] without the need for fixing and staining. SXT captures cellular substructures in their native state, with rapid data collection (<10 min per cell), allowing analysis of relatively large cell populations [16]. This imaging modality operates within the “water window” energy range [17,18], where carbon-rich materials such as membranes and organelles exhibit higher contrast due to stronger X-ray attenuation compared with the cytosol. This quantitative absorption measure, known as linear absorption coefficient (LAC), facilitates the identification of organelles based on their molecular density and composition [19,20]. This unique aspect of SXT allows for the mapping of subcellular organization, as well as subtle fluctuations in the biochemical composition of organelles during different cellular states without the need for labeling [21,22].

Advances in SXT methodology allow us to distinguish α and β cells from each other by quantifying differences in cellular features. Correlated cryogenic fluorescence microscopy and SXT is a powerful approach to confirm the identity of a cell by incorporating a cell-specific fluorescent probe [16]. Furthermore, by analyzing LAC distributions within individual vesicles, we can extract vesicular features that distinguish glucagon and insulin vesicles. These sub-vesicular LAC-based parameters can quantify subtle shifts in biochemical density and the physical organization of secretory vesicle cores. Thus, these parameters can be leveraged to classify islet cells based on the identity of the vesicles within each cell using supervised machine learning classification. Using a predictive model can enable us to systematically quantify cellular classification while avoiding excluding what might otherwise appear to be outlier cells. A similar machine learning approach to identify α and β cells based on their metabolic state has recently been reported [23]; however, this study will be the first cell-type classifier based on 3D ultrastructural information. The most important features distinguishing insulin and glucagon vesicles in these predictive models can be identified and visualized using dimensional reduction techniques such as Uniform Manifold Approximation and Projection (UMAP) [24]. These results will not only clarify key parameters informing our models but also advance our understanding of the biophysical makeup of insulin and glucagon vesicles.

To test our approach, we focused on imaging α and β cells, which are the most abundant cells in mouse islets and challenging to distinguish without careful analysis. We imaged eight α and seven β cells and quantified cell volumes, nuclear occupancy, number of vesicles, and organelle LAC values. Using these subcellular and sub-vesicular features, our machine learning-based approach allows us to differentiate α and β cells with a relatively high accuracy of 77%. Whole-cell metrics can also be embedded in UMAP to distinguish α and β cells from each other. Furthermore, UMAP embeddings allow us to visualize two separate pools of glucagon vesicles and insulin vesicles. This approach enables the determination of systematic and microscopic features of two types of mouse endocrine cells, facilitating future label-free identification of cell types in various species, including humans.

## 2. Materials and Methods

### 2.1. Animal Model

All animal experiments were approved by the Institutional Animal Care and Use Committee at the University of Southern California (Animal Use Protocol #21120). All experiments were performed in accordance with relevant guidelines and regulations. Human proinsulin with C-peptide-bearing Super folder Green Fluorescent Protein (CpepSfGFP) has been expressed in transgenic mice as described previously [25] and maintained within a C57BL6/J strain. This mouse model was provided to our lab by Peter Arvan (University of Michigan). Experiments were performed on adult mice of approximately 2 months of age.

### 2.2. Islet Isolation and Dissociation

Islets were isolated using a 3.5 mg/mL liberase and 1.5 mg/mL DNAse enzyme blend (Roche Diagnostics, Indianapolis, IN, USA) as previously described [26]. Briefly, the enzyme solution was perfused into the pancreas via the bile duct, and the inflated pancreas was removed for further digestion at 37 °C for 12 min. Extracted islets were hand-picked under a fluorescence light microscope to isolate C-peptide-bearing Superfolder Green Fluorescent Protein (CpepSfGFP)-labeled β cell containing islets and incubated with Roswell Park Memorial Institute (RPMI) 1640 (Gibco, New York, NY, USA) at 37 °C overnight before being used for experiments. After overnight incubation, the islets were transported on ice (Integrated Islet Distribution Program, 2020) to the National Center for X-ray Tomography at the Advanced Light Source (Lawrence Berkeley National Laboratory, Berkeley, CA, USA) where the islet dissociation was performed. The CpepSfGFP fluorescence signal was monitored after both the transport and overnight incubation steps to ensure that the signal remained robust. Approximately 300 islets were collected into a microtube, centrifuged at 200 g for 5 min, washed with 1× phosphate buffered solution (PBS) without Ca^2+^ and Mg^2+^ (Gibco) (referred to as 1× PBS in subsequent sections), pelleted again at 200 g for 5 min and then incubated with 700 μL of Accumax Cell Dissociation Solution (Innovative Cell Technologies Inc., San Diego, CA, USA) for 7 min at 37 °C. The dissociated islets were then pelleted at 300 g for 5 min and resuspended in 10 μL of 1× PBS for loading into the microcapillaries. Cell viability was measured to ensure that the dissociated cells were not damaged. The cells were manually counted using trypan blue, yielding a cell viability of ~85%. The process is depicted in Steps 1 and 2 in Appendix A.

### 2.3. Specimen Cryopreservation

The capillaries were pulled in-house with a diameter of ~10 μm to accommodate single cells [27]. The tips of the capillaries have a diameter of 10 μm, which can easily accommodate intact cells and is the ideal size for imaging. Dissociated islets were harvested by centrifugation at 300× *g* for 5 min and subsequently resuspended in 10 μL of 1× PBS and kept on ice to maintain a stable environment before and during the loading. Cells were loaded into thin-wall glass capillaries using a micro-loading tip (1.3 μL per loading) (Step 3, Appendix A). Each capillary was then rapidly plunged into liquid nitrogen-cooled liquid propane and stored in liquid nitrogen until image acquisition [27] (Appendix A).

### 2.4. Fluorescence Microscopy

Prior to dissociation and capillary loading, islets were evaluated for a strong GFP signal. The islets were transferred into an 8-well imaging dish with a #1.5 polymer coverslip (ibidi) containing 1× PBS. A Leica MICA microscope was used to collect fluorescent and transmitted light images of whole islets using the high contrast plan apochromatic (HC PL APO) CS2 63×/1.20 water immersion lens. Auto-illumination was set at 2.105% for the 488 nm laser to capture the GFP signal across the four internal detectors. The pixel density was set at 2432 × 2032 with a 600 Hz bidirectional scan speed and line averaging at 4 repetitions. The pinhole was set to 1 airy unit (AU). A total Z-stack averaged to ~20 μm was collected in ~400 s. The fluorescence image was then processed on the LASX software (Leica Application Suite X, version 6.2.1.27469) using the Lightning and Thunder setting to deconvolve the image with the settings set for adaptive strategy, a refractive index of 1.33, and water as the mounting media. A brightfield image of the same islet was taken at 108 intensity, 40.0 ms exposure, and 1.0 gain.

To confirm that β cells were present in the imageable region of the capillary after loading, the SfGFP signal in the capillary was measured. The capillaries were secured onto a 1× GS-Holder (Okolab, Pozzuoli, Italy). Fluorescence and transmitted light images of SfGFP-tagged β cells inside the capillary were taken with a HC PL FLUOTAR 10×/0.32 dry objective lens using similar settings as the whole islet imaging. ImageJ was used to create maximum projections of the fluorescence images and overlaid them onto the transmitted light image (Appendix A). Owing to strong intercellular interactions between the different cell types, clusters of 2-3 cells could still be observed after loading them into thin-walled glass capillaries. Larger clusters of cells were retained in the base and middle part of the capillary outside of the imageable region (Appendix A—top and middle panel), whereas smaller clusters (2–3 cells) and fully dissociated cells were identified in the imageable region of the capillary (Appendix A—bottom panel). We observed more α cells than β cells in the tip of the capillary, likely due to α cells being smaller than β cells [12] (Appendix A).

### 2.5. Cryogenic Confocal Fluorescence Microscopy

To definitively distinguish between α and β cells, we traced the CpepSfGFP fluorescence signal along the capillary tip by using a cryogenic confocal fluorescence microscope (Appendix A). Cryogenic fluorescence data of the cells and their corresponding bright field images were collected using a home-built cryogenic fluorescence microscope [28,29,30,31], using a commercial dual spinning disk head (CSU-X1, Yokogawa, Tokyo, Japan) for confocal scanning and detection. A laser at 491 nm was used for the detection of the C-peptide-SfGFP fluorescent signal, which was controlled with an acousto–optical tunable filter using an integrated system (Andor Laser Combiner, model LC-501A). Datasets were collected acquiring a Z-stack of 30–40 μm, with ΔZ = 0.3 μm (Appendix A). Data analysis was performed in Fiji (Version 1.54) [32], and the maximum intensity projections were determined. Lastly, we correlated the fluorescence signal with the X-ray absorption signal collected from the same specimen, allowing us to identify β cells.

### 2.6. Transmission Electron Microscopy

Pancreatic cell ultrastructure was imaged using a FEI Talos F200C G1 microscope (Thermo Scientific™, Waltham, MA, USA) to perform transmission electron microscopy (TEM). Briefly, pancreatic tissue samples were fixed in 4% paraformaldehyde in 1× PBS (Gibco), followed by 2.5% glutaraldehyde and 2% paraformaldehyde in 0.1 M N-2-Hydroxyethylpiperazine-N′-2-Ethanesulfonic Acid (HEPES) (Sigma Aldrich, St. Louis, MA, USA) and postfixed in 1% osmium tetroxide overnight. The fixed samples were stained with 1% uranyl acetate (Ted Pella, Redding, CA, USA) for an hour and dehydrated with an increasing percentage of ethanol solutions. Propylene oxide (PO) was used as a transition fluid and embedded with a medium resin hardness using the Embed 812 kit, which polymerized at 60 °C for a minimum of 18 h. Ultrathin sections (80 nm) were obtained using a Leica UC6 ultramicrotome. Once mounted on grids, sections were treated with 3% H_2_O_2_, then stained with lead citrate followed by uranyl acetate. The stained sections were examined using Talos F200C TEM operated at 80 kV. Images were taken with a mounted Ceta Camera (Appendix A). Materials used were obtained from Electron Microscopy Sciences unless otherwise stated.

### 2.7. Soft X-ray Tomography Data Collection and Reconstruction

Projection images were collected at 517 eV using the soft X-ray microscope XM-2 [27,33] at the National Center for X-ray Tomography (Advanced Light Source synchrotron, at Lawrence Berkeley National Laboratory). The microscope was equipped with a 50 nm resolution objective lens. During data collection, cells were maintained in a stream of helium gas cooled to liquid nitrogen temperatures [34], which allows the collection of projection images while reducing the effects of exposure to radiation. Projection images were collected sequentially around a rotation axis of 180°, with 2° increments. Sudden drifts of the capillary are corrected automatically by the data acquisition software (as detailed in Chen et al. 2022 [27]). An exposure time of 350 ms was used for each acquired projection. The 3D image reconstruction was achieved using an iterative reconstruction method in the software package AREC3D. This method uses a 3D model-based approach to align the projection images before reconstruction, as reported by Parkinson et al. 2012 [35]. The LAC value for each tomogram was calculated by normalizing the intensity value of each pixel by the pixel area. The detailed method is described by Chen et al. 2022 [27]. The operation defines the LAC value of each voxel in the tomogram and is a direct measurement of the carbon-atom concentration in each voxel (Appendix A).

For correlating SXT data with the correspondent fluorescent signal, the position of each cell was marked for each capillary and manually retrieved *a posteriori* during the image post-processing.

### 2.8. SXT Data Segmentation LAC Quantification

Dissociated single cells were manually segmented using Amira 2021.2 (Thermo Scientific™, Waltham, MA, USA). Organelle identification and segmentation were performed as previously described [29,36]. The ACSeg 3D U-net model in the Biomedisa platform was used to generate an initial membrane mask [37,38]. The segmentation of the cell membrane was refined using the “Paintbrush” tool every 5–10 slices and interpolated to generate a 3D mask. The “Paintbrush” tool was also used to segment the nucleus. The LAC value threshold for the cellular components was based on previously published data [29]. Each value of the LAC is defined as the X-ray absorption per μm. Insulin vesicles were identified by their characteristic morphology and high LAC value (0.21 μm^−1^–0.46 μm^−1^) compared with other organelles [36,39]. Due to the similar morphology and a denser packaging of glucagon in α cell vesicles [40], segmentation of glucagon vesicles used a similar strategy, albeit at a varied threshold range (0.24 μm^−1^–0.50 μm^−1^). Based on the morphology and size of both types of vesicles, vesicle clusters bigger than 500 nm were excluded. Similarly, vesicles smaller than 100 nm were excluded to avoid smaller synaptic-like GABA vesicles [41,42]. Mitochondria were segmented based on morphology.

### 2.9. Quantification of Cellular and Subcellular Features and Statistical Analysis

#### 2.9.1. Cellular and Organelle Volume Analysis

The percentage of cell volume occupied by the nucleus and vesicles was calculated by dividing the nuclear and total vesicle volume by the cellular volume, respectively. We defined this parameter as nuclear and vesicle occupancy, respectively. The volume of the cytosol was defined as the difference between the cellular volume and the nuclear volume.

#### 2.9.2. LAC Value Comparison

The mean LAC value for the nucleus, mitochondria, cytosol, and vesicles was plotted for each cell type with the minimum and the maximum value for each column plot representing the standard deviation (Appendix A). The mean value was generated by calculating the mean value of the LAC for all the voxels within a specific semantic label. The mean LAC-based values for secretory vesicles (Appendix A) were plotted by pooling together the LAC values from all vesicles belonging to a cell type.

### 2.10. Statistical Analysis

Statistical analysis was performed using Prism (Version 10.1.2, GraphPad Software, La Jolla, CA, USA). The statistical significance between various parameters was calculated using the unpaired *t*-test with Welch’s correction and the one-way ANOVA test with Bonferroni’s correction. Error bars in column plots are representative of the standard deviation. η^2^ values (effect sizes) were calculated using a custom function in Python (version 3.10.9) that would square the correlation ratio of a given vesicle parameter.

### 2.11. UMAP Projection of Multidimensional Structural Data

UMAP (Uniform Manifold Approximation and Projection) plots were created in Python on Jupyter Notebooks using the package umap-learn-0.5.5. Parameters except for skew and kurtosis values were standardized before creating UMAP embeddings. For the UMAP displaying pooled vesicle parameters, the embedding was created using n_neighbors = 50, min_dist = 0.5, and metric = ‘canberra’. For the UMAP of whole cell metrics, the hyperparameters were n_neighbors = 4, min_dist = 0.2, and metric = ‘canberra’. The canberra distance metric has an extra scaling factor compared with default distance metrics, which makes it useful for clustering data points consisting of features from different numerical scales [43,44]. The color scheme used for displaying individual vesicle parameters was inverted ‘cmocean thermal’ from matplotlib.

### 2.12. Machine Learning Modeling and Validation Strategy

Machine learning was performed using a Python (version 3.10.9) environment. Packages were obtained from scikit-learn version 1.4 [45] unless otherwise stated. The machine learning models used were LogisticRegression, RandomForestClassifier, and XGBoost (xgboost v2.0.3) [46]. These were chosen due to their relative ease of training and interpretation, as well as to facilitate a comparative analysis of model performance. Leave One Group Out cross-validation was utilized to ensure the generalizability of these models when predicting vesicle identities in an unseen cell. In this use case, each group refers to all of the vesicles from an individual cell. Machine learning models were evaluated using Accuracy, Precision, Recall, F1 Score, and ROC AOC metrics.

Out of the 7 β cells and 8 α cells used in this study, the vesicles from 6 β cells and 7 α cells were used as the training and validation dataset, while the remaining vesicles from 1 β cell and 1 α cell were used as the test dataset. To reduce the variability in the machine learning modeling, all 56 possible combinations of 1 β cell and 1 α cell left out of 7 β cells and 8 α cells were used to create 56 different models. The average and standard deviation of the evaluation scores, such as the F1 Score and ROC AOC across the 56 models, were then reported. This process was performed each time for Logistic Regression, Random Forest, and XGBoost (Total of 168 models trained). Hyperparameter tuning was performed using Grid Search for Logistic Regression and Random Forest by optimizing the accuracy metric [47]. Values were also standardized prior to machine learning for logistic regression. To train XGBoost models with many hyperparameters, Bayesian Optimization (BayesianOptimization v1.4.3) [48] was used. The Bayesian Optimization search parameters were init_points = 10, n_iter = 15. Specific hyperparameters tuned are listed in Appendix A. All other parameters not listed use the default package implementation.

### 2.13. Interpretation of Feature Importances from Machine Learning

Permutation feature importances were used to rank the relative importance of vesicle parameters. Each column of features was randomly permuted, and then the difference in accuracy score between the original and permuted datasets was used to rank the importance of each feature. To avoid the dilution of the feature importances from multicollinear features, the Pearson correlation coefficient between vesicle parameters was calculated. Then, features were grouped using hierarchical clustering (scipy.cluster.hierarchy v1.13.0) [49] on correlation coefficients. A representative variable from each cluster was then used to display the final feature importances. Final reported feature importances were averaged over the 56 combinations where accuracy was above 75%.

## 3. Results

### 3.1. α and β Cell Morphology Visualized by SXT

We use SXT to generate 3D datasets of single cells, which allows for the direct quantification of the volume of cells and subcellular structures, including the nucleus, mitochondria, vesicles, and the plasma membrane (Figure 1A,B). We collected and analyzed 15 cells in total (7 β cells and 8 α cells). The voxel size of the tomograms ranged from 30 to 45 nm. After the acquisition, reconstruction, and segmentation of the different subcellular structures in each dataset (Figure 1B), we observed a broad range of cell sizes for both α and β cells, with the average cell volume for α cells being 579 ± 247 μm^3^. In contrast, β cells exhibited a significantly larger volume of about 1191 ± 277 μm^3^ (Figure 1C).

We did not observe significant differences in the nuclear volume among different cells (118 ± 42 μm^3^ for β cells versus 112 ± 32 μm^3^ for α cells) (Figure 1D); however, the percentage of cell volume occupied by nuclear volume (nuclear occupancy, see Methods) for β cells (10 ± 3%) was significantly lower compared with α cells (21 ± 5%) (Figure 1E). As for the secretory vesicles of both cell types, we observed a higher number of α cell vesicles per cytosolic volume (3.3 vesicles/μm^3^) compared with β cell vesicles (2 vesicles/μm^3^) (Figure 1F). On measuring secretory vesicle diameter, we observed a significantly higher mean vesicle diameter for α cells (213 ± 21 nm) compared with β cells (163 ± 13 nm, see Figure 1G and Table 1). Comparing the two cell types, we observed a significantly increased LAC value in the α-cell cytosol (0.26 ± 0.02 μm^−1^) compared with β cell cytosol (0.24 ± 0.02 μm^−1^) (Appendix A). In addition, we found a non-significant increase in the LAC value of mitochondria and nuclei in α-cells (0.357 ± 0.03 μm^−1^ and 0.24 ± 0.02 μm^−1^, respectively) compared with β-cell mitochondria (0.335 ± 0.03 μm^−1^) and nuclei (0.21 ± 0.02 μm^−1^) (Appendix A). A major differentiating factor between the two cell types was a significant difference in the LAC values of the vesicles. We observed a significantly higher mean vesicle LAC for α cells (0.375 ± 0.03 μm^−1^) compared with β cells (0.334 ± 0.02 μm^−1^) (Figure 1H).

### 3.2. Analysis of Vesicle Properties in α and β Cells

We performed an in-depth analysis of the dense-core insulin and glucagon vesicle features. To better understand their morphological differences, we pooled the vesicles according to cell type and analyzed differences between the total pooled glucagon (n = 10,964) and insulin (n = 14,960) vesicles. Analyzing pooled vesicles is particularly useful for comparing heterogeneity in vesicle characteristics. This approach involves the scrutiny of tens of thousands of vesicles, offering a more comprehensive description of the vesicular features compared with relying on values derived from averaged cell data (n = 15).

By comparing cumulative pools of insulin and glucagon vesicles, we observed that the average diameters were significantly different (194 nm for glucagon vesicles; 157 nm for insulin vesicles) between the two groups, which reflected the whole cell values reported in Table 1. Pooling vesicles by cell types enables us to not only examine a larger sample size but also allows us to observe the distribution of vesicle sizes. For insulin vesicles, we see around 4.5% of the vesicles under 120 nm, 80% between 120 nm and 180 nm, and 15.5% of the vesicles above 180 nm (Figure 2B, green). Conversely, for glucagon vesicles, we see around 1% of the vesicles under 120 nm, 50% between 120 nm and 180 nm, and 49% of the vesicles above 180 nm (Figure 2B, red).

Along with structural differences, we observed a significant difference in the mean LAC values for both vesicle types. A significantly higher mean LAC value of 0.365 ± 0.04 μm^−1^ was observed for glucagon vesicles compared with 0.328 ± 0.03 μm^−1^ for insulin vesicles (Figure 2C), indicating a significantly higher molecular density for glucagon vesicles. This is in line with EM observations on glucagon granules appearing consistently denser than insulin granules [11,12,40] (Appendix A).

In addition to providing the mean LAC of a vesicle, SXT is also capable of giving us sub-vesicular physical details. Vesicles consist of multiple voxels with unique LAC values, which allow for the mapping of a LAC value distribution for each vesicle. These LAC value histograms can provide in-depth information on how the molecular density is distributed within a vesicle (Figure 3A,B). The LAC value curve for a vesicle can be uniquely described as a combination of the minimum LAC, 25th quantile LAC, mean LAC, 75th quantile LAC, maximum LAC, mode LAC, median LAC, skew, kurtosis, standard deviation, and interquartile distance. When comparing the sub-vesicular parameters between the two vesicle types, we observed that insulin vesicles had significantly different values for all the LAC-based parameters compared with glucagon vesicles (Figure 3C,D and Appendix A, and Table 2). Although the sub-vesicular parameters differed significantly between both types of vesicles, our comparison of effect sizes between the various parameters revealed that maximum vesicle LAC exhibited the highest score when analyzed using a η^2^-test. Conversely, minimum vesicle LAC had the lowest effect size (Appendix A). This could indicate that the maximum vesicle LAC is a more effective parameter differentiating the two vesicle types compared with other LAC-based parameters.

### 3.3. Differentiating α and β Cells Using Machine Learning Models Based on Extracted Vesicle Characteristics

After establishing significant differences between insulin and glucagon vesicles, we created machine learning models that classify the majority vesicle type from a given islet cell and can deduce cellular identity based on this prediction. This machine learning strategy represents an advancement over existing qualitative identification methods by enabling more precise and consistent quantification of cellular classification. Predictive models were trained using individual vesicle features and identity labels as inputs in logistic regression, random forest, and XGBoost-supervised machine learning algorithms. To minimize variability, models were trained for all fifty-six unique combinations of one left out α and one left out β cell from a total of eight α and seven β cells (Figure 4A,B). While overall morphological characteristics such as cell volume differ between α and β cells, incorporating these features into this vesicle-based ML pipeline could bias the model’s predictive performance and complicate the interpretation of its results.

After the machine learning classifiers were trained, the final average accuracies across the three models were higher than or equal to 75% (Table 3). This accuracy will allow us to predict the predominant vesicle type from an islet cell, enabling us to distinguish whether the cell is an α or β cell. The >0.82 receiving operating characteristic area under the curve (ROC AUC) metric across the models confirms that this machine learning strategy results in a high-performance classifier. The average F1 Score, which reflects the rate of incorrectly classified vesicles, was about 0.70 (Table 3). While the F1 Score was relatively lower than other evaluation metrics, this could be due to the partial overlap in the distribution of SXT characteristics between insulin and glucagon vesicles (Figure 2B,C). Therefore, the model performance of machine learning approaches utilizing SXT data could be influenced by the existence of parameter gradients rather than distinct parameter thresholds between two biological categories.

In our machine learning approach, we averaged the performance of 56 fully tuned models with each other. Across these models, even the worst-performing random forest and XGBoost classifiers achieved over 50% accuracy (Appendix A), thereby providing insight into cellular identity. However, a small portion of logistic regression classifiers resulted in accuracies below 50% despite logistic regression having a similar average performance to the other two models. Therefore, future implementations of machine learning to predict cell and vesicle identity would use random forest or XGBoost algorithmic approaches.

### 3.4. Displaying Distinguishing Features of Insulin and Glucagon Vesicles Using Structure-Based UMAP Visualizations

To understand which vesicle features drive the prediction ability of our model, we calculated machine learning feature importance. Permutation feature importances were used as a standardized way to compare importances between models. Additionally, multicollinear features were addressed by performing hierarchical clustering on highly correlated features and picking one representative feature from each cluster to use as a proxy for a group of features (Appendix A). Similar to trends from statistical significance and effect size tests, vesicle LAC mean, LAC standard deviation, and diameter permutation importances were the most predictive of insulin or glucagon vesicle identity (Figure 5A). Although the overall trends in feature importance were similar between all three machine learning models, the relative importance of each parameter was generally lower in logistic regression compared with Random Forest and XGBoost. This discrepancy may be related to an inherent bias of the logistic regression models since some model combinations were underfitting the data. Interestingly, vesicle LAC Skew was also relatively important, particularly in the XGBoost model. This example illustrates how predictive importance and statistical significance from effect size do not always coincide with each other.

While important distinguishing features have been established between insulin and glucagon vesicles, direct visualization and interpretation of pooled vesicles is difficult. Therefore, we performed dimensional reduction using uniform manifold approximation and projection (UMAP) on characteristics from individual vesicles. UMAP has traditionally been used for omics analysis, but recent papers use this algorithm to analyze diverse biological data types [50,51]. In the UMAP plot, high-dimensional vesicle data are represented in 2D space, where each point in the plot represents 1 of 25,384 vesicles. Upon initial inspection of the UMAP embedding, insulin and glucagon vesicles form semi-distinct regions (Figure 5B). These lobular regions illustrate how insulin and glucagon vesicles originate from different distributions, as demonstrated with our SXT data. The overlap between lobes of vesicle populations further clarifies the performance of machine learning models. While the majority of vesicles are accurately classified, some vesicles in this overlap region are not accurately classified. When vesicles are colored based on cellular origin, insulin vesicles seem to be well distributed throughout the high dimensional space (Figure 5C). This suggests that trends in vesicle metrics are cell-independent. A similar pattern is also observed for glucagon vesicles, with the exception of vesicles from α_3, which tend to have a lower LAC mean. While α_3 has similar whole cell metrics to other α cells, such as overall average vesicle diameter and cell volume (Appendix A), it has a lower average vesicle LAC. As shown by a UMAP plot based on whole cell metrics of α and β cells instead of individual vesicles (Appendix A), it is likely that this finding is reflective of structural cellular heterogeneity within α cells. α cell transcriptional heterogeneity has been previously reported [52], but our results provide evidence of structural diversity as well.

Differentiating characteristics can be interpreted on the UMAP plot by coloring each vesicle according to its vesicle feature values. When vesicles are colored by LAC mean, a gradual and global transition from high to low LAC mean values can be observed in the UMAP plot (Figure 5D). The top left lobe is composed of glucagon vesicles with high LAC, while the top right lobe has primarily insulin vesicles with low LAC, which is consistent with our previous findings. However, there is significant heterogeneity in the bottom lobes, suggesting that subpopulations of vesicles are present within SXT images of α and β cells. In contrast, the LAC standard deviation and diameter plots primarily reflect overall differences between insulin and glucagon vesicles. Higher LAC standard deviation and diameter values correspond to the regions dominated by glucagon vesicles (Figure 5D). Interestingly, there is also a middle-right cluster composed of higher-diameter insulin vesicles in the UMAP space. These vesicles have low LAC mean and LAC maximum (Appendix A), which indicates the presence of a subpopulation composed of immature vesicles despite their relatively larger size. While LAC standard deviation and diameter mainly discriminate vesicle types, the trends in the skew and kurtosis plots do not seem to reflect different insulin or glucagon vesicle UMAP clusters. However, a subgroup of glucagon vesicles seems to be characterized by a relatively lower LAC skew, which could explain this parameter’s significance in the feature importance calculations.

## 4. Discussion

This study establishes the feasibility of leveraging subcellular features to distinguish between two similar secretory islet cells, α and β cells, using SXT and supervised machine learning-based classification algorithms. Our mapping approach provided quantitative differences in the cellular and vesicular structure, providing new insights into the unique biophysical characteristics of the two systems. Given a similar nuclear volume and a significantly larger cellular volume of β cells compared with α cells, we observe a significantly higher nuclear occupancy and a higher number of vesicles per cytosolic volume for α cells. These structural trends for both cell types are in line with previous work performed on visualizing α and β cells quantitatively using serial block face scanning electron microscopy (SBF-SEM) and focused ion beam scanning electron microscopy (FIB-SEM) [8,9,10,11].

The unique ability of SXT to quantify differences in the molecular density of the vesicles in both cell types revealed that glucagon vesicles have a significantly higher average molecular density compared with insulin vesicles. These observations corroborate previous electron microscopy data, reporting glucagon vesicles being more electron-dense [11,12,40]. While EM studies reported insulin vesicles in the range of ~10,000 per cell [8,9,10], SXT captured fewer vesicles (~2000). This is expected, as it was recently shown that SXT does not allow us to visualize immature vesicles without well-defined dense cores [10,36,53].

In contrast to EM studies, we observed a larger vesicle diameter for glucagon vesicles (194 nm) in α cells compared with insulin vesicles (157 nm) in β cells [8,10]. Our results of vesicle diameters we obtained are more in line with the core diameters of insulin and glucagon vesicles (~200 nm). Previous work on measuring core diameters has been performed on small datasets (~400 vesicles) using chemically fixed islets [8]. Our method employs cryo-fixation as opposed to chemical fixation, preserving the subcellular environment and cell morphology in its native state. Thus, the slight differences we observe between SXT and EM measurements of dense cores could be due to differences in the sample preparation.

In addition, due to the relatively large number of vesicles we quantify with SXT (~25,000) compared with EM (~400), we expect that our data can provide a complete description of the dense core size distributions, incorporating cell-to-cell heterogeneity. Moreover, analyzing the entire volume of the cell provides an accurate description of the vesicle-to-vesicle variability within a single cell.

The use of volumetric and biochemical information from the two cell types also helps to rule out the presence of additional islet cell types, such as δ and pancreatic polypeptide (PP) cells. Based on previous studies [11,12], δ cells present a dendrite-like shape and contain lozenge-shaped granule cores with diameters spanning from 200 to 350 nm. These granules have been reported to be less dense compared with glucagon granules [11,12,40]. ε and PP cell secretory vesicles have a much smaller vesicle diameter, which spans from 100 to 150 nm [11,54,55,56,57], which we do not observe in our cells. Moreover, the percentage of δ cells, PP cells, and ε cells in rodent islets is 1–5%, ~1%, and <1% of the whole cellular population, respectively. This is a very low proportion of cells compared with α and β cells (20% and 70%, respectively) [2,58,59]. Therefore, there is a low chance that δ cells, PP cells, and ε cells will be found in the microcapillary tip. Nonetheless, further development of the method could provide the possibility to map volumetric 3D features of additional islet cell types.

SXT also allows us to rigorously quantify the biochemical and biophysical organization of vesicles by defining a LAC value histogram for each vesicle. To perform this analysis, we pooled vesicles from each cell type. We report significant differences in all the LAC-based parameters between insulin and glucagon vesicles (Figure 3). Out of all the differences in these parameters, we observed the maximum LAC and minimum LAC to have the highest and lowest effect sizes, respectively, indicating that the maximum LAC is a more effective differentiating factor between vesicle types compared with minimum LAC. Previous work performed on modeling insulin vesicles using SXT data has shown that the LAC value reduces radially from the center of the vesicles [53]. In line with this, we qualitatively observed maximum LAC voxels residing on the inside of both types of vesicles and the minimum LAC voxels being on the periphery. Future work could focus on differences in the radial distribution of LAC values between vesicle types to better understand how glucagon and insulin are stored in dense cores.

By leveraging the high number of vesicles and vesicle parameters in our pooled vesicle analysis, we were able to differentiate the two cell types based on vesicle identity prediction. Machine learning classifiers were trained on insulin and glucagon vesicles extracted from thirteen cells and tested on two random cells. Out of the three algorithms trained, we obtained the best accuracy (77%) in identifying the vesicles from an unseen cell using a random forest model. While XGBoost is often perceived as the best predictive algorithm for tabular data, other work describes similar performance between random forest and XGBoost in biological binary classification [60,61]. Additionally, machine learning approaches using cellular data recently reported similar accuracy to our own results [23,61,62], demonstrating that meaningful classifications can still be made in the diverse cellular milieu. Similar to hypothesis testing results, the most important differentiating vesicle features were mean LAC, LAC standard deviation, and diameter. While LAC skew and kurtosis were less important, including them in our ML analysis did not seem to affect performance significantly (Appendix A) and allowed us to quantify their overall differentiating power. Our approach demonstrates the capability of SXT to extract generalizable features that can be used to discriminate between different types of secretory vesicles and successively differentiate cells.

Furthermore, SXT presents an innovative method to track biologically meaningful structural heterogeneity in the cell. Both α cells and β cells have been reported to be phenotypically heterogeneous with distinct subtypes that are important for islet function [10,52,63,64]. However, most of these studies investigate transcriptional and functional differences with little quantification of structural features. By embedding cellular parameters in UMAP, we were able to visualize α and β cells in higher dimensional structural space (Appendix A). Although we have not fully sampled all possible structural subtypes of islet cells, we can still report two distinct clusters of cells based on cellular identity. Future studies with a larger number of cells would allow for the identification of distinct subpopulations of islet cells with varied structural features.

We also observe a heterogeneous mixture of vesicles within single cells. Using UMAP embeddings to visualize insulin and glucagon vesicles presents a novel method to not only distinguish two types of vesicles but also categorize and identify subpopulations within larger vesicle pools (Figure 5B). In the future, unbiased graph-based clustering methods could be used to detect vesicle communities and their shifts under the effect of different pharmacological stimulations. Collectively, our findings demonstrate the utility of SXT in investigating heterogeneity in a variety of cell and vesicle types. Overall, 3D studies will help to understand and correlate cellular structural features (i.e., phenotypes) to their physiological role in maintaining cell-cell communication and inform about islet function [65].

Future use of this approach will enable the analysis of pancreatic islets from multiple species, which often cannot be easily genetically manipulated to perform correlated imaging studies. Mapping ultrastructural rearrangements in the 3D structure of the cell is a necessary step in understanding the evolution and progression of disease. SXT can reveal the spatiotemporal evolution of subcellular architecture by informing relationships between organelle spatial distribution and pathological alterations of cellular function. These whole-cell datasets will provide unique input data for integrative whole-cell models [66,67,68,69]. Future studies that quantify mitochondrial networks in β cells may provide new insights into metabolic processes in health and disease in different model organisms.

## 5. Conclusions

In conclusion, we present a soft X-ray tomography and machine learning-based pipeline to classify primary mouse α and β cells and quantify their subcellular structural characteristics. We report significant differences in the ultrastructure of both cell types and their secretory vesicles. Using these subcellular features, we present a generalizable method to robustly quantify differences among and within cell types and vesicles. This approach enables us to investigate biological heterogeneity at a subcellular level and can be extended to other cellular systems.

## Figures and Tables

**Figure 1 cells-13-00869-f001:**
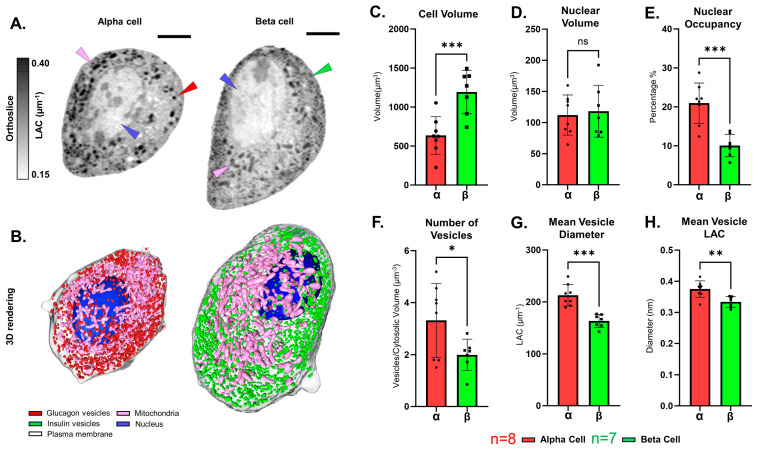
3D reconstruction and quantitative analysis of α and β cell morphology. (**A**) Orthoslice showing the XY plane through the soft X-ray tomogram of representative α and β cells (α_3 and β_6, respectively). Cell constituents and organelles are distinguished from one another based on their LAC values and are identified as follows: nucleus–blue arrowhead, mitochondria–pink arrowhead, glucagon vesicles–red arrowhead, and insulin vesicles–green arrowhead. The overall LAC value range of the orthoslice is between 0.15 and 0.4 μm^−1^ to optimize contrast. Scale bar: 2 μm. (**B**) 3D reconstruction of the representative α and β cells (α_3 and β_6, respectively). In detail, the reconstruction shows the nucleus (blue), mitochondria (pink), glucagon vesicles ((**left**), in red), insulin vesicles ((**right**), in green), and plasma membrane (gray). (**C**) Cellular volume of both cell types, showing a significantly higher volume (*** *p* < 0.001) for β cells (1191 ± 277 μm^3^) compared with α cells (579 ± 247 μm^3^). (**D**) Nuclear volume of both cell types showed no significant difference (*p* = 0.76). (**E**) Comparison between mean nuclear occupancy for each cell type, with a significant increase (*** *p* < 0.001) in percentage occupancy of the nucleus for α cells (21 ± 5%) compared with β cells (10 ± 3%). (**F**) Number of insulin vesicles normalized by cytosolic volume indicating a significantly higher number of vesicles (* *p* = 0.03) per cytosolic μm^3^ for α cells (3.3 ± 1.4 vesicles/μm^3^) compared with β cells (2 ± 0.6 vesicles/μm^3^). (**G**) Plot of mean vesicle diameters of α and β cell vesicles demonstrating a higher vesicle diameter (*** *p* < 0.001) for α cell vesicles (212 ± 21 nm) compared with β cell vesicles (163 ± 13 nm). (**H**) Mean Vesicle LAC for secretory vesicles of α and β cells showing a significantly higher mean LAC (** *p* = 0.003) for α cell vesicles (0.37 ± 0.03 μm^−1^) compared with β cell vesicles (0.33 ± 0.02 μm^−1^). Error bars in each plot are representative of the standard deviation. Welch’s *t*-test was used as a statistical test. n = 8 for α cells (red) and n = 7 for β cells (green).

**Figure 2 cells-13-00869-f002:**
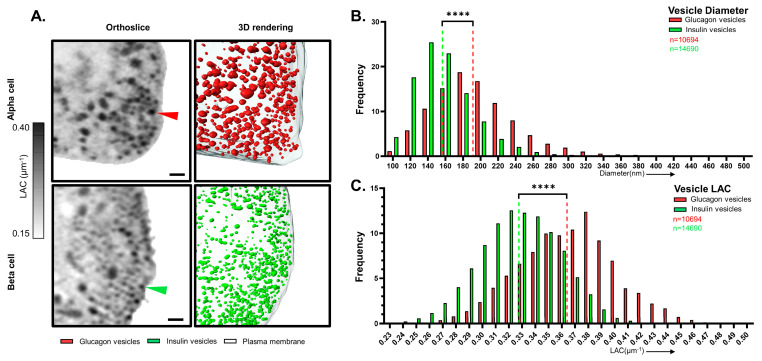
3D reconstruction and quantitative analysis of pooled insulin and glucagon vesicles. (**A**) (**left**) XY Orthoslice through the SXT of representative α and β cells. Glucagon vesicles (red arrowheads) and insulin vesicles (green arrowheads) can be identified based on their high LAC values. The overall LAC value in the orthoslice is thresholded between 0.15 and 0.40 μm^−1^ (scale bar: 0.5 μm). (**right**) 3D reconstruction of a section of representative α and β cells (α_3 and β_6, respectively). In detail, the reconstruction shows glucagon vesicles ((**top**), in red) and insulin vesicles ((**bottom**), in green), and plasma membrane (gray). (**B**) Histogram showing the size distribution of glucagon and insulin vesicles. The vesicles for each cell type are pooled together and show a significantly higher diameter (**** *p* < 0.0001) for insulin vesicles (194.2 ± 49 nm, green dotted line), compared with glucagon vesicles (157 ± 35 nm, red dotted line). (**C**) Histogram showing LAC distribution of glucagon and insulin vesicles demonstrating a significantly higher mean vesicle LAC values (**** *p* < 0.0001) for insulin vesicles (0.37 ± 0.04 μm^−1^, red dotted line), compared with glucagon vesicles (0.33 ± 0.03 μm^−1^, green dotted line). (**B**,**C**) n = 10,694 for glucagon vesicles (red) and n = 14,690 for insulin vesicles (green). Welch’s *t*-test was used as a statistical test.

**Figure 3 cells-13-00869-f003:**
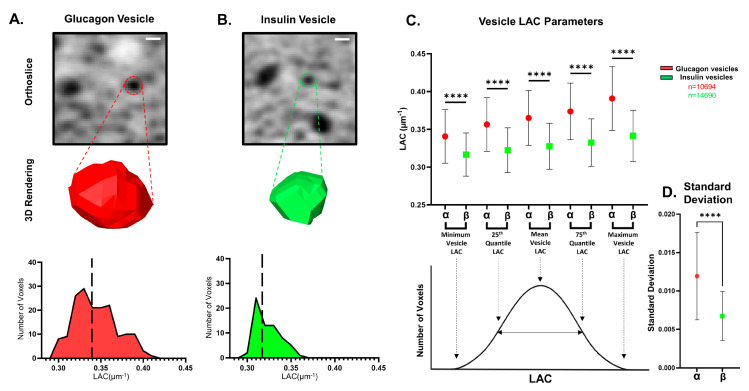
Description and comparison of LAC-based parameters between insulin and glucagon vesicles (**A**) (**top**) XY Orthoslice through SXT of representative α cell (α_3). Scale bar: 0.5 μm. (**middle**) 3D reconstruction of a representative glucagon vesicle (red). (**bottom**) Histogram displaying the LAC distribution of the glucagon vesicle picture in the top and middle panels showing a mean LAC value of 0.34 μm^−1^ for the vesicle. (**B**) (**top**) XY Orthoslice through SXT of representative β-cell (β_6). Scale bar: 500 nm. (**middle**) 3D reconstruction of a representative insulin vesicle (green). (**bottom**) Histogram displaying the LAC distribution of the insulin vesicle picture in the (**top**) and (**middle**) panels, showing a mean LAC value of 0.318 μm^−1^ for the vesicle. (**C**) (**top**) A comparison of vesicle LAC parameters (minimum LAC, 25th quantile LAC, mean LAC, 75th quantile LAC, maximum LAC) between glucagon vesicles (red) and insulin vesicles (green) showing significantly higher values (**** *p* < 0.0001; one-way ANOVA with Bonferroni’s correction) for glucagon vesicles for all displayed parameters compared with insulin vesicles. (**bottom**) LAC histogram curve for a sample vesicle, with arrows indicating the value being compared in the (**top**) panel. (**D**) Plot showing a significantly higher (**** *p* < 0.0001; Welch’s *t*-test) standard deviation for glucagon vesicles (red) compared with insulin vesicles (green). Error bars in all plots are representative of the standard deviation. n = 10,694 for glucagon vesicles (red) and n = 14,690 for insulin vesicles (green).

**Figure 4 cells-13-00869-f004:**
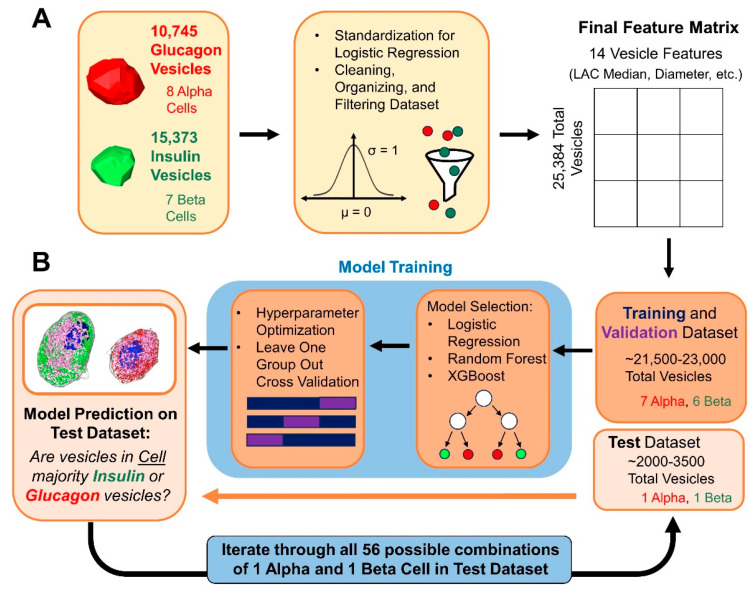
Overview of machine learning strategy. (**A**) Data pre-processing for insulin and glucagon vesicles from β and α cells is conducted. A final vesicle feature matrix, including group labels (denoting which cell a vesicle is from) for vesicles, is used as input for machine learning. (**B**) Train/test split for grouping vesicles from α and β cells. The process of model building is described, with Leave One Group Out cross-validation used to estimate the performance of predicting vesicle identity from unseen cells. Model building and testing are repeated over 56 combinations to understand variability in performance.

**Figure 5 cells-13-00869-f005:**
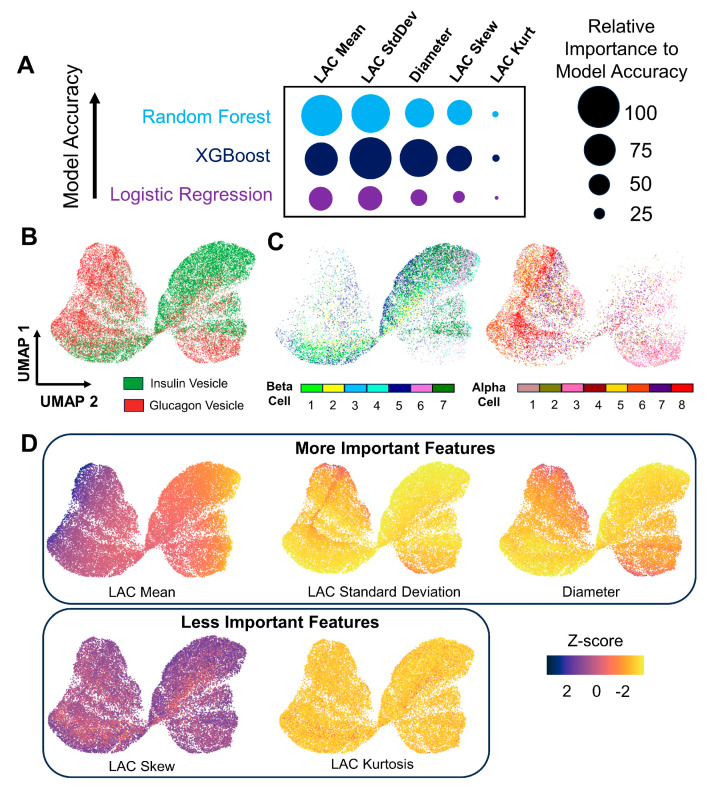
Representing vesicle feature importances in UMAP embeddings. (**A**) Representative feature importances from each ML model listed in order of accuracy. The radius of circles is scaled to the magnitude of the permutation feature importances. Since the LAC standard deviation in the XGBoost model had the highest overall importance magnitude, the other parameters are scaled to it. In general, LAC mean, LAC standard deviation, and diameter seem to be the most important representative parameters. (**B**) UMAP embedding of vesicles colored by vesicle identity. Semi-distinct clusters of insulin and glucagon vesicles can be observed. (**C**) Vesicles colored by cellular origin. The overall trends in the pooled vesicle UMAP space do not seem to be driven by cell-dependent effects. (**D**) Embeddings colored by vesicle feature values. Gradients of LAC mean, standard deviation, and diameter correspond to regions of insulin and glucagon vesicles. The grouping of heterogeneous vesicle subpopulations can also be visualized.

**Table 1 cells-13-00869-t001:** Cumulative analysis of the segmented organelles in α and β cells. List of the overall volume, occupancy (%), and LAC values for the cell, nucleus, mitochondria, and vesicles in each cell type, along with the number of vesicles and vesicle diameter (nm). Each value is reported with its standard deviation. Overall, we observe that β cells have a higher cell volume, similar nuclear volume, lower nuclear occupancy, mean vesicle diameter, and mean vesicle LAC compared with α cells.

Cell Type	α-Cell	β-Cell
Number of Cells	8	7
Cell Volume (µm^3^)	579 ± 247	1191 ± 277 ***
Nucleus Volume (µm^3^)	112 ± 32	118 ± 42
Nucleus Volume (%)	21 ± 5	10 ± 3 ***
Vesicle Volume (µm^3^)	6 ± 2	5 ± 2
Vesicle Volume (%)	1.1 ± 0.4	0.4 ± 0.1
Vesicle Number	1337 ± 480	2099 ± 710 *
Vesicle Diameter (nm)	213 ± 21	163 ± 13 ***
Vesicle LAC (µm^−1^)	0.375 ± 0.03	0.334 ± 0.02 **
Nucleus LAC (µm^−1^)	0.24 ± 0.02	0.21 ± 0.02
Mitochondria LAC (µm^−1^)	0.357 ± 0.03	0.335 ± 0.03
Cytosol LAC (µm^−1^)	0.263 ± 0.02	0.237 ± 0.02 *

Data were analyzed using Welch’s *t*-test: * *p* < 0.05, ** *p* < 0.01, *** *p* < 0.001.

**Table 2 cells-13-00869-t002:** Cumulative analysis of the pooled insulin and glucagon vesicles. List of the average mean, median, mode, maximum, minimum, 25th quantile, and 75th quantile LAC along with average number of vesicles, standard deviation, skewness, kurtosis, and diameter in both vesicle types. Each value is reported with its standard deviation.

Vesicle Type	Glucagon Vesicle	Insulin Vesicle
Number of Vesicles	10,964	14,960
Mean LAC (µm^−1^)	0.365 ± 0.04	0.328 ± 0.03 ^†^
Median LAC (µm^−1^)	0.364 ± 0.04	0.327 ± 0.03 ^†^
Mode LAC (µm^−1^)	0.365 ± 0.04	0.328 ± 0.03 ^†^
Maximum LAC (µm^−1^)	0.391 ± 0.04	0.342 ± 0.03 ^†^
Minimum LAC (µm^−1^)	0.341 ± 0.04	0.317 ± 0.03 ^†^
Standard Deviation (µm^−1^)	0.012 ± 0.005	0.007 ± 0.003 ^†^
25th Quantile LAC (µm^−1^)	0.356 ± 0.04	0.322 ± 0.03 ^†^
75th Quantile LAC (µm^−1^)	0.374 ± 0.04	0.332 ± 0.03 ^†^
Skewness	0.085 ± 0.37	0.288 ± 0.44 ^†^
Kurtosis	−0.506 ± 0.49	−0.617 ± 0.55 ^†^
Diameter (nm)	194 ± 49	157 ± 35 ^†^

Data were analyzed using Welch’s *t*-test; ^†^
*p* < 0.0001.

**Table 3 cells-13-00869-t003:** Final evaluation metrics from supervised machine learning strategy for classifying cells based on vesicle predictions. Accuracy and other evaluation metrics for the three models are reported as a mean and standard deviation across the 56 model combinations. Based on accuracy and AUC metrics, the machine learning model predicts the majority of vesicles in cells with significant statistical power. F1 scores indicate that a portion of the vesicles are misclassified.

Model	Accuracy	F1 Score	ROC AUC
Logistic Regression	0.75 ± 0.11	0.68 ± 0.12	0.82 ± 0.13
Random Forest	0.77 ± 0.11	0.71 ± 0.09	0.85 ± 0.10
XGBoost	0.75 ± 0.12	0.70 ± 0.11	0.83 ± 0.11

## Data Availability

Reconstructed data are available from the corresponding contacts upon request. The original code is available at the GitHub repository: https://github.com/kvn42999/Subcellular-Feature-Based-Classification-Alpha-Beta-Cells.git (accessed on 9 April 2024.

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
