# Peer review of "Subcellular Feature-Based Classification of α and β Cells Using Soft X-ray Tomography"

_cells, 2024, doi:10.3390/cells13100869_

Round 1

Reviewer 1 Report

Comments and Suggestions for Authors

The authors developed a subcellular feature-based classification approach to distinguish cryo SXT images of alpha and beta cells from mouse pancreatic islet tissue and quantify the subcellular characteristics of both cell types.

The authors describe significant differences in the cell morphology and organelle distributions, specifically in the sizes and density distributions of glucagon and insulin vesicles. The paper develops a supervised machine learning approach to predict the identity of cell types, which, in principle, can be generally applied to other systems to identify cell subtypes from cryo SXT imaging. As such the paper is interesting and provides not only useful insights into the pancreatic alpha and beta cell biology, but also provides a generally applicable toolset for label-free identification of cell subtypes and study their cell heterogeneity. The paper is technically sound, well written. I recommend it for publication.

A few minor points.

The machine learning model is based on a variety of vesicle features and achieves an accuracy of ~75%. I wonder why the authors did not combine information about the global cell morphology together with vesicle-based features. The cell volume seems to be dramatically different in alpha and beta cells. I wonder if this addition could have improved the classification even further. The authors don’t need to redo the analysis, but they could comment on this point.

There is a large variance in cell volumes for both alpha and beta cells. Some cells are twice as large as others in the same subtype. I wonder if the authors found any substantial differences in cell morphology or structural features that mirrors the large volumetric differences.   

It seems that LAC Skew and LAC Kurtosis do not quite contribute to machine learning training. Would it be better if they are not included?

Overall, the paper enhances our understanding of structural differences between alpha and beta cells as well as the biophysical makeup of insulin and glucagon vesicles. I recommend the manuscript for publication.

Author Response

Subcellular Feature-Based Classification of α and β Cells using Soft X-ray Tomography

Manuscript ID: cells-3008688

Thank you very much for taking the time to review this manuscript. Please find the detailed responses below and the corresponding revisions highlighted in yellow in the resubmitted files.

Response to Reviewer 1 (black text is reviewer comment, blue text is our response)

Reviewer 1:

The authors developed a subcellular feature-based classification approach to distinguish cryo SXT images of alpha and beta cells from mouse pancreatic islet tissue and quantify the subcellular characteristics of both cell types.

The authors describe significant differences in the cell morphology and organelle distributions, specifically in the sizes and density distributions of glucagon and insulin vesicles. The paper develops a supervised machine learning approach to predict the identity of cell types, which, in principle, can be generally applied to other systems to identify cell subtypes from cryo SXT imaging. As such the paper is interesting and provides not only useful insights into the pancreatic alpha and beta cell biology, but also provides a generally applicable toolset for label-free identification of cell subtypes and study their cell heterogeneity. The paper is technically sound, well written. I recommend it for publication.

Comments:

  1. The machine learning model is based on a variety of vesicle features and achieves an accuracy of ~75%. I wonder why the authors did not combine information about the global cell morphology together with vesicle-based features. The cell volume seems to be dramatically different in alpha and beta cells. I wonder if this addition could have improved the classification even further. The authors don’t need to redo the analysis, but they could comment on this point.

We thank the reviewer for this comment, and have added this sentence to section 3.3:

“While overall morphological characteristics like cell volume differ between α and β cells, incorporating these features into this vesicle-based ML pipeline could bias the model’s predictive performance and complicate interpretation of its results.” Page 11 Line 435 - 438.

An issue with incorporating overall morphological features in the machine learning pipeline is interpreting features that are measured 15 times (cell volume) with features that are measured 25,384 times (vesicles). Also, the interpretation of vesicle feature importances in section 3.4 would be difficult if a non-vesicle feature like cell volume was included in the ML classifiers. However, this is an interesting idea for us to consider in our future developments.

  1. There is a large variance in cell volumes for both alpha and beta cells. Some cells are twice as large as others in the same subtype. I wonder if the authors found any substantial differences in cell morphology or structural features that mirrors the large volumetric differences.

We thank the reviewer for pointing this out.

We observed concurrent changes in nuclear volumes and numbers of vesicles along with an increase in cell size. This is reflected in the fact that some nuclei from the same subset of cells are double the size as the others as well. We also think that similar changes might be reflected in other organelles such the ER and mitochondria. As our method focuses mainly on quantifying vesicles, future efforts to visualize other organelles might provide more information on the heterogeneity of structural features within a cell type.

  1. It seems that LAC Skew and LAC Kurtosis do not quite contribute to machine learning training. Would it be better if they are not included?

We thank the reviewer for pointing this out. We have added the following to the Discussion section:

“While LAC skew and kurtosis were less important, including them in our ML analysis did not seem to affect performance significantly (Figure S6E) and allowed us to quantify their overall differentiating power.” Page 15 Line 611 - 614.

Re-training ML models by excluding LAC skew and kurtosis values did not seem to change performance significantly, as demonstrated by an example now included in Figure S6E. Also, including LAC skew and kurtosis values in the feature importance calculations allows us to compare importance values with results from effect size and hypothesis testing.

Overall, the paper enhances our understanding of structural differences between alpha and beta cells as well as the biophysical makeup of insulin and glucagon vesicles. I recommend the manuscript for publication.

Reviewer 2 Report

Comments and Suggestions for Authors

The authors do not state precisely the method used for the SXT reconstruction, which is crucial for calculating the LAC. Furthermore, in acquisition during rotation, movements of the specimen can occur which then affect the quality of the reconstructed tomography. Have the authors taken this phenomenon into account? In addition I suggest to declare the voxel size of the reconstructed tomography.

I suggest reversing panels G and H in figure 1 because in section 3.1, panel H (diameter) is described before panel G (LAC).

I suggest the same reversion also in figure 2 because panel C and D are described before A

In the description of the panel C and B of figure 2 the authors do not mention in the text that the distribution of vesicle diameter and LAC are statistically different. Please insert in the text as well.

In section 2.8 the authors should briefly explain how LAC is calculated

I suggest inserting figure S6 in the main text, to facilitate understanding of the machine learning method used.

Page 11 line 440-443 This is not clear please explain better the meaning of these phrases

Page 12 lines 461-463. Please clarify how the authors derive the features importance shown in fig 4A

In Figure 4c it is better, in the panel of Beta cells, shows only the points of this type of cells. The same for the alfa. Do not plot in grey the point of the other type of cells.

Page 12 line 482 The authors declare that some vesicles are not accurately classified. Which are these vesicles?

Page 14 line 550. The author discussed the motivation why they reveal a larger vesicles diameter than EM. Among them I suggest to mention the difference in the samples preparation and that the acquisition was made in cryo, conditions that preserve better the subcellular organelles and in general the whole morphology of the cells.

Page 15 line 585 An accuracy of 77% is not so high, so I suggest to discuss better this point and compare with others similar papers.

Page 15 line 599-601 It is not clear the meaning of this phrase. Please try to explain better.

Page 1 line 29-30 (Abstract) Please re-phrase

In section 2.10 define eta-square as effect size.

Table 2 is never cited in the main text

To improve the

Panel D of figure 3 is never cited in the main text

Page 4 line 175 Figure 2C is not correct

Page 4 line 192 in the Figure S3A in the supplementary Letter A is not present in the figure.

Page 5 line 202 “350 ms was used for the entire acquisition”. Is the exposure for a single tomogram? it is not clear written like this

page 5 line 205 in the Figure S3B in the supplementary Letter B is not present in the figure

Author Response

Subcellular Feature-Based Classification of α and β Cells using Soft X-ray Tomography

Manuscript ID: cells-3008688

Thank you very much for taking the time to review this manuscript. Please find the detailed responses below and the corresponding revisions highlighted in yellow in the resubmitted files.

Response to Reviewer 2 (black text is reviewer comment, blue text is our response)

Reviewer 2:

Comments:

  1. The authors do not state precisely the method used for the SXT reconstruction, which is crucial for calculating the LAC. Furthermore, in acquisition during rotation, movements of the specimen can occur which then affect the quality of the reconstructed tomography. Have the authors taken this phenomenon into account?

We thank the reviewer for this comment.  For data alignment and reconstruction on the X-ray microscope XM-2, we used standard operative methods described in previous works published by the National Center for X-ray Tomography. To clarify the methods used for data reconstruction and capillary realignment, we have added the following sentences:

“Sudden drifts of the capillary are corrected automatically by a software integrated in the data acquisition suit (as detailed in Chen et al. 2022  [27]).” Page 5, line 201-203.

And also:

“The 3D image reconstruction was achieved using an iterative reconstruction method in the software package AREC3D. This method uses a 3D model-based approach to align the projection images before reconstruction, as reported in Parkinson et al. 2012 [35]. The LAC value for each tomogram was calculated by normalizing the intensity value of each pixel by the pixel area. The detailed method is described by Chen et al. 2022 [27]. The operation defines the LAC value of each voxel in the tomogram and is a direct measurement of the carbon-atom concentration in each voxel (Figure S3).” Page 5 Line 205-210.

  1. In addition I suggest to declare the voxel size of the reconstructed tomography.

We thank the reviewer for this comment.  We have added the following sentence:

“The voxel size of the tomograms ranged from 30–45 nm.” Page 7 Line 301-302.

  1. I suggest reversing panels G and H in figure 1 because in section 3.1, panel H (diameter) is described before panel G (LAC).

I suggest the same reversion also in figure 2 because panel C and D are described before A

We thank the reviewer for this comment.

Panel G and H in Fig 1. have been interchanged.

  1. In the description of the panel C and B of figure 2 the authors do not mention in the text that the distribution of vesicle diameter and LAC are statistically different. Please insert in the text as well.

We thank the reviewer for this comment. We have inserted the following into two sentences:

“By comparing cumulative pools of insulin and glucagon vesicles, we observed that the average diameters were significantly different (194 nm for glucagon vesicles; 157 nm for insulin vesicles) between the two groups which reflected the whole cell values reported in Table 1.” Page 8 Line 358-361.

“A significantly higher mean LAC value of 0.365±0.04 μm-1 was observed for glucagon vesicles compared to 0.328±0.03 μm-1 for insulin vesicles (Figure 2C), indicating a significantly higher molecular density for glucagon vesicles.”  Page 9 Line 382-384

  1. In section 2.8 the authors should briefly explain how LAC is calculated

We thank the reviewer for this comment.  To clarify how the LAC was calculated, we have added the following sentences:

 “Each value of the LAC is defined as the X-ray absorption per μm.” Page 5, line 221-222:

 “The LAC value for each tomogram was calculated by normalizing the intensity value of each pixel by the pixel area. The detailed method is described by Chen et al, 2022 [27]. The operation defines the LAC value of each voxel in the tomogram and is a direct measurement of the carbon-atom concentration in each voxel (Figure S3).” Page 5, line 205-210:

  1. I suggest inserting figure S6 in the main text, to facilitate understanding of the machine learning method used.

We thank the reviewer for this comment. Figure S6 has been inserted into the main text as Figure 4.

  1. Page 11 line 440-443 This is not clear please explain better the meaning of these phrases

We thank the reviewer for this comment. We have changed the sentence into the following:

“In our machine learning approach, we averaged the performance of 56 fully tuned models with each other. Across these models, even the worst performing random forest and XGBoost classifiers achieved over 50% accuracy (Figure S6D), thereby providing insight into cellular identity.” page 12 line 459 - 462.

  1. Page 12 lines 461-463. Please clarify how the authors derive the features importance shown in fig 4A

We thank the reviewer for this comment. We have changed the sentence to include the following:

“Similar to trends from statistical significance and effect size tests, vesicle LAC mean, LAC standard deviation, and diameter permutation importances were the most predictive of insulin or glucagon vesicle identity (Figure 5A).” Page 12 Line 479 - 482.

  1. In Figure 4c it is better, in the panel of Beta cells, shows only the points of this type of cells. The same for the alfa. Do not plot in grey the point of the other type of cells.

We thank the reviewer for this comment. We have changed Figure 5 as described in this feedback.

  1. Page 12 line 482 The authors declare that some vesicles are not accurately classified. Which are these vesicles?

We thank the reviewer for this comment. We have changed the sentence to include the following:

“While the majority of vesicles are accurately classified, some likely in this overlap region are not.” Page 13 Line 501 - 502.

  1. Page 14 line 550. The author discussed the motivation why they reveal a larger vesicles diameter than EM. Among them I suggest to mention the difference in the samples preparation and that the acquisition was made in cryo, conditions that preserve better the subcellular organelles and in general the whole morphology of the cells.

We thank the reviewer for this comment. We have changed the sentence to include the following:

“Our method employs cryo-fixation as opposed to chemical fixation, preserving the subcellular environment and cell morphology in its native state. Thus, the slight differences we observe between SXT and EM measurements of dense cores could be due to differences in the sample preparation.” Page 15 Line 566-569

  1. Page 15 line 585 An accuracy of 77% is not so high, so I suggest to discuss better this point and compare with others similar papers.

We thank the reviewer for this insight. We have changed the section to include the following:

“Additionally, machine learning approaches using cellular data recently report similar accuracy to our own results [23, 61,62], demonstrating that meaningful classifications can still be made in the diverse cellular milieu.” Page 15 Line 607 - 610.

References 23 and 62 use metabolic autofluorescence signatures and transcriptomics data respectively to achieve an accuracy of 65-85% using supervised machine learning to classify α and β cells . Reference 61 also achieves an accuracy of 65-75% using supervised machine learning to classify sex and age based on multi-omics data.

  1. Page 15 line 599-601 It is not clear the meaning of this phrase. Please try to explain better.

We thank the reviewer for this comment. We have changed the sentence to the following:

“Although we have not fully sampled all possible structural subtypes of islet cells, we can still report two distinct clusters of cells based on cellular identity.”  page 16 lines 623 - 624.

  1. Page 1 line 29-30 (Abstract) Please re-phrase

We thank the reviewer for this comment. We have changed the sentence to the following:

“These sub-vesicular parameters enable us to systematically predict cell type using supervised machine learning.”  Page 1 Line 29 - 30

  1. In section 2.10 define eta-square as effect size.

We thank the reviewer for this comment. We have changed the sentence to include the following:

“η2 values (effect sizes) were calculated using a custom function in Python (version 3.10.9) that would square the correlation ratio of a given vesicle parameter.” Page 6 Line 248 - 249.

  1. Table 2 is never cited in the main text

We thank the reviewer for this comment. We have cited it in the following sentence:

“When comparing the sub-vesicular parameters between the two vesicle types, we observed that insulin vesicles had significantly different values for all the LAC based parameters as compared to glucagon vesicles (Figure 3C, Figure 3D, Figure S5A,B and Table 2).” Page 9 Line 394-397.

  1. Panel D of figure 3 is never cited in the main text

We thank the reviewer for this comment. We have cited it in the following sentence:

“When comparing the sub-vesicular parameters between the two vesicle types, we observed that insulin vesicles had significantly different values for all the LAC based parameters as compared to glucagon vesicles (Figure 3C, Figure 3D, Figure S5A,B and Table 2).” Page 9 Line 394-397.

  1. Page 4 line 175 Figure 2C is not correct

We thank the reviewer for this comment. We have changed the sentence to the following:

“Datasets were collected acquiring a Z-stack of 30 – 40 μm, with ΔZ=0.3 μm (Figure S2C).” Page 4 Line 174-175

  1. Page 4 line 192 in the Figure S3A in the supplementary Letter A is not present in the figure.

We thank the reviewer for this comment. We have changed the sentence to the following:

“Images were taken with a mounted Ceta Camera (Figure S3).” Page 4 Line 191-192

  1. Page 5 line 202 “350 ms was used for the entire acquisition”. Is the exposure for a single tomogram? it is not clear written like this

We thank the reviewer for this comment. We have changed the sentence to the following:

“An exposure time of 350 ms was used for each projection.” Page 5 Line 203-204.

  1. page 5 line 205 in the Figure S3B in the supplementary Letter B is not present in the figure

We thank the reviewer for this comment. We have changed the sentence to the following:

“The operation defines the LAC value of each voxel in the tomogram and is a direct measurement of the carbon-atom concentration in each voxel (Figure S3).” Page 5 Line 210.